# Combined *Phyllostachys pubescens* and *Scutellaria baicalensis* Prevent High-Fat Diet-Induced Obesity via Upregulating Thermogenesis and Energy Expenditure by UCP1 in Male C57BL/6J Mice

**DOI:** 10.3390/nu14030446

**Published:** 2022-01-19

**Authors:** Yoon-Young Sung, Seung-Hyung Kim, Dong-Seon Kim

**Affiliations:** 1KM Science Research Division, Korea Institute of Oriental Medicine, 1672 Yuseong-daero, Yuseong-gu, Dajeon 34054, Korea; yysung@kiom.re.kr; 2Institute of Traditional Medicine and Bioscience, Daejeon University, 62 Daehak-ro, Dong-gu, Daejeon 34520, Korea; sksh518@gmail.com

**Keywords:** adipogenesis, AMPK, brown adipose tissue, FNDC5, PGC-1α, SIRT1

## Abstract

This study examined the anti-obesity effects of a *Phyllostachys pubescens* (leaf) and *Scutellaria baicalensis* root mixture (BS21), and its underlying mechanisms of action, in high-fat diet (HFD)-induced obese mice. Mice were fed a HFD with BS21 (100, 200, or 400 mg/kg) for 9 weeks. BS21 reduced body weight, white adipose tissue (WAT) and liver weights, liver lipid accumulation, and adipocyte size. Additionally, BS21 reduced serum concentrations of non-esterified fatty acid, triglyceride, glucose, lactate dehydrogenase, low-density lipoprotein cholesterol, total cholesterol, leptin, and insulin growth factor 1, but elevated the adiponectin concentrations. Furthermore, BS21 suppressed the mRNA levels of lipogenesis-related proteins, such as peroxisome proliferator–activated receptor (PPAR) γ, SREBP-1c, C/EBP-α, fatty acid synthase, and leptin, but increased the mRNA gene expression of lipolysis-related proteins, such as PPAR-α, uncoupling protein (UCP) 2, adiponectin, and CPT1b, in WAT. In addition, BS21 increased the cold-stimulated adaptive thermogenesis and UCP1 protein expression with AMPK activation in adipose tissue. Furthermore, BS21 increased the WAT and mRNA expression of energy metabolism-related proteins SIRT1, PGC-1α, and FNDC5/irisin in the quadriceps femoris muscle. These results suggest that BS21 exerts anti-obesity and antihyperlipidemic activities in HFD-induced obese mice by increasing the thermogenesis and energy expenditure, and regulating lipid metabolism. Therefore, BS21 could be useful for preventing and treating obesity and its related metabolic diseases.

## 1. Introduction

Obesity is defined by excessive body fat mass due to an imbalance between energy intake and expenditure [1] and is a common health problem that increases the risks for other metabolic disorders, such as dyslipidemia, insulin resistance, hypertension, cancer, and cardiovascular disorders [2]. White adipose tissue (WAT) is the primary storage organ for excess energy, primarily in the form of triglycerides, whereas brown adipose tissue (BAT) contains a large number of mitochondria that disperse stored energy (fat) as heat [3]. Brown adipocytes uncouple the mitochondrial electron transport chain from ATP synthesis by permeabilizing the mitochondrial inner membrane to allow intermembrane protons to leak back into the mitochondrial matrix through the mitochondrial uncoupling protein (UCP) 1, consequently reducing ATP synthesis and generating heat [4]. Male animals maintained at thermoneutrality with UCP1 deficiency had elevated weight gain and glucose intolerance [5]. Activation of UCP1 by pharmacological drugs has been shown to suppress obesity and improve insulin sensitivity through its thermogenic function [6].

Pharmacotherapies for obesity are currently used in clinical practice; however, these weight loss drugs induce significant side effects that often limit their use [7,8]. Thus, natural herbal medicines have recently emerged as alternatives that can overcome these limitations [9]. In Korean and Chinese traditional medicine, *Phyllostachys pubescens* leaves have been used to treat hypertension and cerebral palsy, and they have various biological activities, including antioxidant and anticoagulant activities [10,11]. *Scutellaria baicalensis* roots have been used as medicinal plants for diuretic, antidiarrheal, and anti-inflammatory effects [12]. Recently, the various flavonoids found in *P. pubescens* leaves and *S. baicalensis* root were shown to exert anti-obesity and antihyperlipidemic effects [13,14]. In our previous study, we selected several plants to screen for anti-obesity effects among some of the plants known as safe to consume. *P. pubescens* leaves and *S. baicalensis* root showed the most reliable anti-obesity activites among the plants. We previously observed that diverse combinations of *P. pubescens* leaves and *S. baicalensis* roots (BS) have anti-obesity and antihyperlipidemic activities in high-fat diet (HFD)-fed obese rodents; in particular, a 2:1 mixture (BS21) was the most effective ratio for anti-obesity activity [15]. BS21 inhibited the adipocyte differentiation in 3T3-L1 adipocytes by downregulating lipogenic or adipogenic proteins, such as CCAAT/enhancer-binding protein-α (C/EBP-α), peroxisome proliferator–activated receptor γ (PPAR-γ), and sterol regulatory element-binding protein 1c (SREBP-1c); and inducing browning marker protein UCP1 [16]. However, the effects and underlying mechanisms of BS21 on obese animals have not been clearly investigated. Thus, this study explored the anti-obesity and antihyperlipidemic activities of BS21, and the underlying mechanism on obesity in HFD-fed obese animals. This study was also conducted to determine the appropriate dose prior to clinical trials for commercialization of standardized BS21.

## 2. Materials and Methods

### 2.1. Extraction

*P. pubescens* leaves and *S. baicalensis* roots as dried plants were obtained from Zhenjiang KOC (Jiangsu, China). The two plants were prepared by extraction with ethanol (70%) as solvent for 3 h, then the mixture (BS21) was mixed with a 2:1 ratio of *P. pubescens* and *S. baicalensis* as concentrate and dried. The powder extracts were stored at −20 °C for further study. From Ultra Performance Liquid Chromatography analysis for chemical profiling, we identified that BS21 contains baicalin, baicalein, orientin, chlorogenic acid, chrysin, tricin, wogonin, and wogonoside [16].

### 2.2. Obesity Induction and BS21 Administration in Animals

For obesity induction, male C57BL/6J mice were obtained from Daehan BioLink (Eumsung, Korea) at 6 weeks of age (average body weight, 18 g). The animals were adapted under temperature (23 ± 3 °C) and humidity-controlled conditions with a 12-h/12-h light/dark cycle. After adaption with standard rodent chow diet for 1 week, the animals were assigned to one of 6 groups matched for body weight: (1) the normal diet (ND) group, mice fed with standard chow diet containing 11.5% fat, 20.8% protein, and 67.7% carbohydrates (*n* = 10); (2) the HFD group, mice fed with an HFD containing 60% fat, 20% protein, and 20% carbohydrates (Rodent Diet D12492, Research Diets, New Brunswick, NJ, USA, *n* = 10); (3) the HCA group, HFD plus *Garcinia Cambogia* extract containing hydroxycitric acid (HCA) (Novarex, Cheongju, Korea) at 245 mg/kg, which was based on the human dosage (HCA 245 mg/kg, *n* = 10); (4) the 100 mg/kg BS21 group, HFD plus BS21 extract at 100 mg/kg (*n* = 10); (5) the 200 mg/kg BS21 group, HFD plus BS21 extract at 200 mg/kg (*n* = 10); and (6) the 400 mg/kg BS21 group, HFD plus BS21 extract at 400 mg/kg (*n* = 10). In previous studies, the anti-obesity effects of BS21 mixtures were investigated in 100 mg/kg dose using a HFD obesity mice [15]. This study was conducted to determine the appropriate dose prior to clinical trials for commercialization of BS21. Thus, BS21 dose was set at three doses to investigate the effect of each dose based on the single dose (100 mg/kg) in the preliminary experiment. For a positive control, *Garcinia cambogia* containing HCA was used because it is known as a popular weight loss dietary supplement (14). ND and HFD groups only received the vehicle. BS21 and HCA were mixed with vehicle solution (0.5% carboxyl methyl cellulose) and treated by oral gavage once daily with the HFD. Food intake and body weight were obtained once per week, and food efficiency ratio (FER) was determined as follows: (total weight gain/total food intake) × 100. These studies were approved by the Institutional Animal Care and Use Committee of Daejeon University (Approval No. DJUARB2020-013), and all animal studies were conducted in accordance with the National Research Council’s Guide for the Care and Use of Laboratory Animals.

### 2.3. Collection of Blood, Liver, Muscle, and Brown or White Adipose Tissues

Blood samples from mice were collected via cardiac puncture under anesthesia after fasting. Serum were determined by centrifugation at 3000× *g* for 20 min at 4 °C. Following blood collection, the liver, quadriceps femoris muscle, BAT, and WAT (epididymal, retroperitoneal, intestinal, and inguinal WAT) were excised immediately, weighed, and stored for analysis.

### 2.4. Serum Biochemical Analyses and ELISA

Serum total cholesterol, non-esterified fatty acid (NEFA), high-density lipoprotein (HDL)-cholesterol, low-density lipoprotein (LDL)-cholesterol, glucose, creatinine, aspartate aminotransferase (AST), alanine aminotransferase (ALT), triglyceride (TG), and lactate dehydrogenase (LDH) concentrations were evaluated by a biochemical analyzer (Hitachi-7020, Hitachi Medical, Japan). Leptin (#MOB00), adiponectin (#MRP300), and insulin growth factor (IGF) 1 (#MG100) hormone concentrations from serum were determined using mouse enzyme-linked immunosorbent assay (ELISA) kits (R&D Systems, Minneapolis, MN, USA).

### 2.5. Liver TG Level Measurement

TG and protein levels from the homogenate of liver tissue were determined from the glycerol values using the Triglyceride assay kit (#ab65336, Abcam, Cambridge, UK) and DC protein assay (Bio-Rad, Hercules, CA, USA), respectively. Briefly, liver tissue was homogenized in 5% NP-40 solution and the homogenate was centrifuged for 2 min. The samples were mixed with assay buffer and lipase, and incubated for 20 min for convert TG to glycerol and fatty acid. Additionally, TG reaction mix was added into each wells and incubated for 60 min. Absorbance was measured on microplate reader at OD 570 nm. The TG levels were normalized to protein concentration for each sample.

### 2.6. Histological Analyses of Liver and Adipose Tissues

For histology, BAT, WAT, and liver tissue were fixed and then embedded in paraffin. All tissues sectioned (4-μm thickness) and dyed to hematoxylin and eosin (H&E). Adipocyte size from epididymal WAT was measured in an area of 20 adipocytes in a stained section per mouse; data represent the mean of 10 animals [13]. The cells were counted and measured with a light microscope (200× *g* magnification, Olympus, Tokyo, Japan).

### 2.7. RNA Isolation and Real-Time PCR Analysis

Total RNA from EWAT and quadriceps femoris muscle was isolated by Trizol solution (Sigma-Aldrich, St. Louis, MO, USA) and translated to cDNA using the FirstStrand cDNA synthesis kit (Amersham Pharmacia, Philadelphia, PA, USA). The mRNA expression levels were evaluated using gene-specific primers and probe, and TaqMan Gene Expression and Power SYBR Green PCR Master Mix (Applied Biosystems, Foster City, CA, USA), respectively, in ABI PRISM 7700 systems. PCR reactions were performed according to the following protocol: samples were heated to 94 °C for 2 min, followed by 40 cycles of 94 °C for 20 s, 60 °C for 20 s, and 72 °C for 30 s. The expression levels of the genes of interest in each treated group were determined after normalizing levels to *Gapdh* expression as an internal control by using comparative cycle threshold (Ct) analysis. The used primer and probe sequences are shown in Appendix A.

### 2.8. Cold Tolerance Test

The animals were transferred to a cold chamber at an ambient temperature (4 °C) for the cold tolerance test. The rectal temperature during the cold tolerance test was measured using a TH-8 Thermalert Monitoring Thermometer (Physitemp, Clifton, NJ, USA) for 3 h in the cold chamber. The mice were sedated and restrained for < 30 s during the measurement. BS21 samples were orally administered to mice and the rectal temperature was determined at 0, 30, 60, 90, 120, and 180 min after BS21 treatment.

### 2.9. Immunofluorescence Staining of UCP1

The BAT sections were fixed, permeabilized, blocked with 5% bovine serum album for 16 h at room temperature, and incubated with UCP1 primary antibody (#ab10983, Abcam, Cambridge, UK) for overnight at 4 °C, washed three times with 1× phosphate-buffered saline solution containing Tween 20 and treated with fluorophore-labeled secondary antibody (#A10040, Invitrogen, Carlsbad, CA, USA) for 2 h. For nuclear staining, section was incubated with Hoechst 33,258 (2.5 µg/mL, bis-benzimide, Sigma, St. Louis, MO, USA) for 10 min. Sections were observed using an Eclipse Ti-E inverted fluorescent microscope (200× magnification, Nikon Instruments Inc., Mississauga, ON, Canada).

### 2.10. Western Blot Analysis

Protein extraction from the epididymal WAT was prepared using Intron Pro-prep solution (Intron, Seoul, Korea). The protein concentrations from each tissue were obtained using a detergent compatible (DC) colorimetric assay (Bio-Rad, Hercules, CA, USA), and the protein expression was determined using the primary antibodies of p-AMPK (#2535, Clone 40H9), AMPK (#5831, clone D5A2), UCP1 (#14670, clone D9D6X), and β-actin (#8457, clone D6A8) (Cell Signaling, Beverly, MA, USA), respectively. The bands were exposed by enhanced chemiluminescence solution using a LAS 4000 (GE Healthcare Life Sciences, Chicago, IL, USA). The expression density of the visualized protein band was obtained using Image J1.49 software. The target protein concentrations were normalized to β-actin as the internal control.

### 2.11. Statistical Analysis

A difference of *p* < 0.05 was considered statistically significant. The significant differences among groups were determined by one-way analysis of variance followed by Dunnett’s multiple comparison test using Prism 7.0 software. Data are presented as the means ± standard error of the mean (SEM).

## 3. Results

### 3.1. Effects of BS21 on Body Weight, Body Weight Gain, Food Intake, and FER

The body weight, body weight gain, food intake, and FER of the animals are shown in Figure 1. The body weight increased with an HFD, and the weight gain was higher in the HFD group than in the ND group (*p* < 0.01) (Figure 1A). However, mice in all BS21 groups and the HCA group (positive control) had a significantly lower average body weight and significantly less body weight gain than the HFD (Figure 1A–C). Among the groups, food intake was not significantly different (Figure 1D). The HFD mice had a higher FER than those in the ND group, all BS21 groups, and the HCA group (*p* < 0.05) (Figure 1E). Visual observation demonstrated a marked difference between groups in the size of mice (Figure 1F).

### 3.2. Effects of BS21 on Serum Biochemical Parameters

The effects of BS21 on biochemical serum parameters are shown in Figure 2. Triglyceride, total cholesterol, LDL-cholesterol, glucose, NEFA, and LDH levels were higher in the HFD mice than in the ND mice (*p* < 0.001, except glucose levels) and significantly lower in all BS21-group mice than HFD-group mice (*p* < 0.05) (Figure 2A–C,E–G). No differences in HDL-cholesterol concentrations were found among the groups (Figure 2D). Creatinine concentrations were reduced in the 200 mg/kg BS21 mice compared with the HFD mice (*p* < 0.01) (Figure 2H). The HFD mice had higher serum ALT than the mice in all the BS21 groups (*p* < 0.001), although no differences in AST concentrations were found among the groups (Figure 2I,J).

### 3.3. Serum Leptin, Adiponectin, and Insulin Growth Factor (IGF) 1 Levels

Serum leptin levels were elevated in the HFD mice but remained low in the BS21 mice (*p* < 0.05) (Figure 3A). By contrast, adiponectin concentrations were higher in the 200 and 400 mg/kg BS21 mice than in the HFD mice (*p* < 0.05) (Figure 3B). Serum IGF-1 concentrations were elevated in the HFD group (*p* < 0.001) but inhibited in the HCA (positive control) and 400 mg/kg BS21 groups (*p* < 0.05) (Figure 3C).

### 3.4. Adipose Tissue Weight

As shown in Figure 4A, the HFD mice had greater subcutaneous inguinal fat (IWAT), visceral fat (intestinal WAT (INWAT), retroperitoneal WAT (RWAT), and epididymal WAT (EWAT)) percentages and liver weights than those in the ND group (*p* < 0.001). EWAT, RWAT, INWAT, and the visceral fat total percentage were reduced in the BS21 groups (*p* < 0.05) (Figure 4B,C). Subcutaneous IWAT was also decreased in the BS21 groups. The total body fat (sum of subcutaneous IWAT and visceral WAT) percentage was reduced significantly in BS21 mice (*p* < 0.05) (Figure 4D). The HFD mice had increased liver weights and liver fat (TG) accumulation; these levels were lower in the HCA positive control and all BS21 groups (*p* < 0.05) (Figure 4E,F).

### 3.5. Histologic Examination

Histologic examination of the H&E-stained liver tissues revealed many lipid droplets in the HFD-group mice, but not in the ND-group mice (Figure 5A). This increased liver fat accumulation (steatosis) induced by the HFD was decreased in mice that received BS21. Increased adipose tissue weight can be due to hypertrophy or hyperplasia of adipocytes, or a combination of both. Histologic examination of IWAT and EWAT tissue sections by H&E staining showed larger adipocytes in the HFD group than in the ND group (Figure 5B,C). By contrast, adipocytes were considerably smaller in all BS21 mice and the HCA control group. The 200 and 400 mg/kg BS21 mice showed fewer large adipocytes (size, > 30 μm) and more small adipocytes (size, 10~30 μm) than the HFD group (Figure 5D). The 400 mg/kg BS21 mice had significantly fewer large adipocytes (size, 90~110 μm or 110~130 μm) than HFD mice (*p* < 0.05).

### 3.6. Effects of BS21 on Lipogenesis or Lipolysis-Related Gene Expression in EWAT

The HFD group showed higher mRNA expression levels of lipogenesis-associated proteins C/EBP-α (*Cebpa*), FAS (*Fasn*), SREBP-1c (*Srebf1*), PPAR-γ (*Pparg*), and leptin (*Lep*) than the ND group (*p* < 0.001) (Figure 6A). Expression of these lipogenesis-related genes was significantly reduced in the BS21 group (200 and 400 mg/kg) and HCA control (*p* < 0.05) (Figure 6A). Transcription of lipolysis-related proteins, such as PPAR-α (*Ppara*), UCP-2 (*Ucp2*), adiponectin (*Adipoq*), and CPT1b (*Cpt1b*), was significantly increased in the BS21 (100, 200, and 400 mg/kg) groups (*p* < 0.01) (Figure 6A).

### 3.7. Energy Expenditure and Thermogenesis

Cold-stimulated adaptive thermogenesis was assessed to investigate its association with energy expenditure and contribution to the suppression of body weight gain [17]. The BS21 mice showed higher average rectal temperatures under cold exposure for 30 and 60 min (*p* < 0.01 and *p* < 0.05, respectively) than the HFD mice (Figure 7A). In addition, the protein levels of the thermogenic protein UCP1 from immunofluorescence analysis were increased in the BAT of all BS21-group mice, but did not increase in the control-group mice (Figure 7B). Furthermore, the protein expression of UCP1 with p-AMPK in EWAT and IWAT significantly increased in the BS21 (200 and 400 mg/kg) mice (*p* < 0.05) (Figure 7C,D).

In addition, the quadriceps femois muscle weight was increased in the BS21 groups and significantly increased by the treatment of BS21 (400 mg/kg, *p* < 0.05) (Figure 8A). The mRNA expression of energy expenditure and thermogenesis-related proteins Silent mating type information regulation 2 homolog 1 (Sirtuin, SIRT1) (*Sirt1*), peroxisome proliferator–activated receptor γ coactivator (PGC)-1α (*Ppargc1a*) and fibronectin type III domain-containing protein (FNDC) 5/irisin (*Fndc5*) in the muscle were markedly induced in the BS21 groups (*p* < 0.05) (Figure 8B).

## 4. Discussion

Our previous study identified BS21, the most effective ratio of a combination of *P. pubescens* leaf and *S. baicalensis* root extract for protection against obesity. The study examined the effects of BS21 on obesity to uncover its underlying mechanisms of action. Our studies showed that BS21 reduced body weight, body weight gain, FER, fat weight, and adipocyte size. Specifically, BS21 was more effective than the positive control HCA at reducing body weight and fat weight. Reduced body weight gain by BS21 could be attributable to reduced fat weight independent of food consumption because the food intake was unchanged across groups and the FER was lower in BS21-treated mice than in the HFD mice. These results indicate that BS21 inhibits lipid accumulation in adipose tissue and liver tissue, which may attenuate adiposity. These observations are consistent with our previous study showing the anti-obesity effects of various BS mixtures; however, the effects and underlying mechanisms of BS21 from previous study, especially on energy metabolism, have not been clearly explored [15].

Adipocytes secrete various bioactive substances known as adipokines such as adiponectin and leptin. Leptin regulates food intake and energy expenditure, whereas adiponectin regulates lipid and glucose metabolism and can suppress the development of metabolic disorders. Leptin and adiponectin concentrations have been shown to be positively and negatively associated with obesity, respectively [18,19]. This study found that BS21 decreased serum leptin and elevated adiponectin concentrations, which is supported by the reduced adipocyte size in WAT. These observations suggest that the regulation of circulating adipokines by BS21 is attributable for suppressing adipocyte size and lipid accumulation in fat tissue.

Obesity elevates the uptake of serum free fatty acids into the liver and fatty acid synthesis, which can lead to fatty liver [20]. Moreover, obesity is often accompanied by an altered serum lipid profile. BS21 decreased weight and TG accumulation from liver and also reduced serum TG, NEFA, cholesterol, and LDL-cholesterol concentrations in HFD mice. Our findings suggest that BS21 can efficiently regulate lipid metabolism, indicating that it may improve fatty liver and hyperlipidemia. In addition, BS21 lowered the circulating glucose and IGF-1 levels which were beforehand increased by HFD-induced obesity, thereby suggesting that BS21 can regulate insulin resistance. Normally, TG induction in the liver causes dysfunction of and damage to the liver. BS21 reduced serum liver function markers such as ALT and LDH levels but had no effect on the kidney function marker creatinine. This suggests that BS21 improved potential HFD-induced liver damage and does not seem to cause detectable liver and kidney toxicity.

Obesity results from an imbalance between energy intake and expenditure and lipogenesis (adipogenesis) and lipolysis. One putative way to combat obesity is preventing adipogenesis. This would preclude the creation of adipocytes and fat accumulation, which would minimize energy storage and increase energy expenditure. Here, we examined lipid metabolism-related gene expression from adipose tissues to uncover the mechanisms of BS21′s anti-obesity effect. Fat accumulation and adipocyte differentiation have major roles in the development of obesity. C/EBPα, PPAR-γ, and SRBEP-1c as transcription factors play important roles in adipogenesis and induce the synthesis of fatty acids and triglycerides via activating the expression of genes or proteins for adipogenesis and lipogenesis, including fatty acid binding protein 4 and FAS [21,22]. This leads to accelerated lipogenesis for triglyceride formation in the terminal phase of differentiation. BS21 reduced the mRNA expression of adipogenesis/lipogenesis-related genes PPAR-γ (*pparg*), SREBP-1c (*Srebf1*), C/EBPα (*Cebpa*), and FAS (*Fasn*); and increased the mRNA expression of lipolysis-related genes PPAR-α (*Ppara*), UCP2 (*Ucp2*), adiponectin (*Adipoq*), and CPT1b (*Cpt1b*) in WAT of HFD mice. Consistently with these results, in our previous study using 3T3-L1 adipocytes, BS21 inhibited the adipocyte differentiation by downregulating PPAR-γ, SREBP-1c, and C/EBPα [16]. These results indicate that decreased adipogenesis or lipogenesis and increased lipolysis by BS21 administration inhibit body weight and adiposity in HFD-induced obese mice.

Next, to identify the other mechanisms of the inhibitory effect of BS21 on obesity, we examined the stimulatory effects of BS21 on thermogenesis and energy expenditure. Adipose tissue, especially BAT, produces heat through expressing UCP1 by PGC-1α and increasing mitochondrial biogenesis and density [23]. Thus, BAT acts as a major thermogenic tissue to induce energy expenditure by activating key regulators of thermogenesis [24]. In particular, UCP1, a main factor of BAT, defends against cold and obesity. The transformation of WAT into BAT can be an appropriate pharmacological target for increased energy consumption and obesity [25]. From the fluorescence staining result of our study, BS21 administration induced the protein expression of thermogenic UCP1 in the BAT of obese mice. Additionally, BS21 increased the UCP1 and p-AMPK protein expression in WAT. A previous study reported that increased UCP1 expression by AMPK activation induces heat production and energy expenditure through browning of adipose tissue [26]. Consistently with these results, in our previous study using 3T3-L1 adipocytes, BS21 induced the browning marker genes UCP1 and PGC-1α expression with AMPK activation [16]. Baicalin, a main component of the root of *S. baicalensis*, attenuated the HFD-induced obesity through upregulating the expression of thermogenic genes (UCP1 and PGC-1α) in adipocytes [27]. These results indicate that BS21 promotes energy expenditure and alleviates obesity-induced abnormalities in lipid metabolism.

We further identified whether BS21 alleviates HFD-induced biological dysfunction in skeletal muscle. Obesity is closely associated with mitochondrial loss and dysfunction. Obesity-induced muscle mitochondrial changes include reduced muscle mitochondria count and muscle oxidative capacity and decreased expression in a master mitochondrial gene regulating oxidative metabolism [28]. Thus, we investigated mRNA levels of mitochondrial genes such as SIRT1, PGC-1α, and FNDC5 in skeletal muscle. The results showed that BS21 treatment dramatically upregulated the mRNA expression of SIRT1 (*Sirt1*), PGC-1α (*Ppargc1a*), and FNDC5/irisin (*Fndc5*), due to increased muscle weight loss by obesity. Skeletal muscle produces and secretes various myokines to mediate muscle metabolism and maintain body homeostasis [29,30]. Irisin is a mainly produced myokine secreted from skeletal muscle and adipose tissue, and this protein is secreted into the circulation after the cleavage of FNDC5 and PGC-1α. Irisin promotes the browning of WAT and the metabolic rate of the whole body [31]; thus, this myokine can prevent or treat obesity and metabolic diseases [32]. Some studies have reported that PGC-1α overexpression induces browning and UCP1 upregulation in WAT by FDNC5/irisin [33,34,35]. Lipid and glucose metabolism from muscle can be regulated by stimulation of the AMPK-SIRT1-PGC-1α signaling pathway, and the activities of AMPK, SIRT1, and PGC-1α can be inhibited by obesity and other metabolic diseases, which impair lipid and glucose metabolism in skeletal muscle [36,37,38,39]. Our data showed that SIRT1, PGC-1α, and FNDC5/irisin expression levels were increased by BS21 administration in the muscle of HFD-fed mice. Thus, this study indicates that stimulation of the SIRT1-PGC-1α signaling in skeletal muscle has a significant role in mediating the beneficial effects of BS21 on lipid and glucose metabolism of obesity. BS21 could attenuate muscle loss and promote lipid and glucose metabolism partially through the SIRT1-PGC-1α-FNDC5/irisin signaling pathway in muscle. Therefore, these result show that BS21 is a potential therapeutic target for improving obesity.

## 5. Conclusions

Our results show that BS21 reduces obesity and hyperlipidemia induced by an HFD via increasing thermogenesis and exergy expenditure by UCP1. Collectively, our findings suggest that BS21 could be useful for the prevention and treatment of obesity and its related metabolic diseases. Based on the results of this preclinical animal study, we are now progressing to a clinical study of BS21 in humans with obesity.

## Figures and Tables

**Figure 1 nutrients-14-00446-f001:**
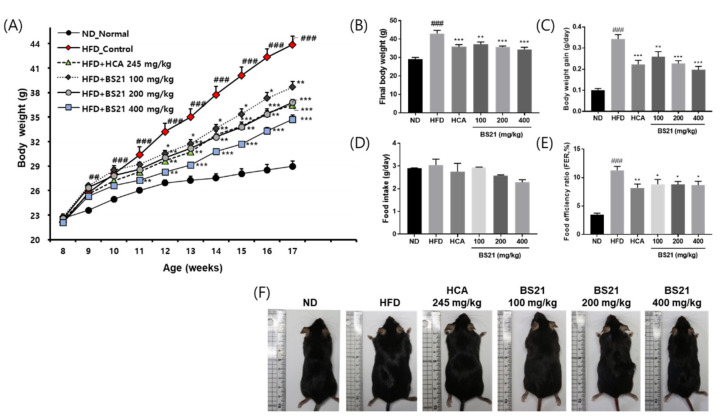
Effects of BS21 on body weight, body weight gain, food intake, and food efficiency ratio in high-fat diet (HFD)-induced obese mice. (**A**) Body weight change. (**B**) Final body weight by group. (**C**) Body weight gain by group. (**D**) Food intake by group. (**E**) Food efficiency ratio by group. (**F**) A representative image of each mouse group. BS21, mixture of *Phyllostachys pubescens* leaf and *Scutellaria baicalensis* root extracts at a ratio of 2:1; HCA, *Garcinia cambogia* extract containing hydroxycitric acid (HCA); ND, normal diet. Values are means ± SEM (*n* = 10). ## *p* < 0.05 and ### *p* < 0.001 vs. the ND; * *p* < 0.05, ** *p* < 0.01 and *** *p* < 0.001 vs. the HFD.

**Figure 2 nutrients-14-00446-f002:**
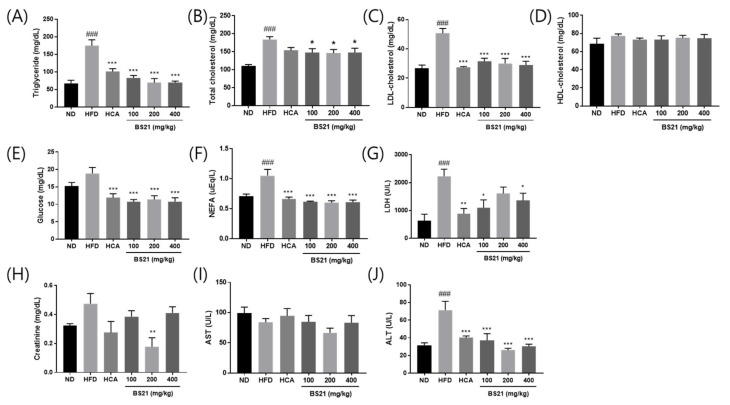
Effects of BS21 on serum biochemical markers in high-fat diet (HFD)-induced obese mice. (**A**) Triglyceride, (**B**) total cholesterol, (**C**) LDL-cholesterol, (**D**) HDL-cholesterol, (**E**) glucose, (**F**) non-esterified fatty acid (NEFA), (**G**) lactate dehydrogenase (LDH), (**H**) creatinine, (**I**) AST, and (**J**) ALT levels. BS21, HFD + mixture of *Phyllostachys pubescens* leaf and *Scutellaria baicalensis* root extracts at a ratio of 2:1; HCA, HFD + *Garcinia cambogia* extract; ND, normal diet. Values are means ± SEM (*n* = 10). ### *p* < 0.001 vs. the ND; * *p* < 0.05, ** *p* < 0.01 and *** *p* < 0.001 vs. the HFD.

**Figure 3 nutrients-14-00446-f003:**
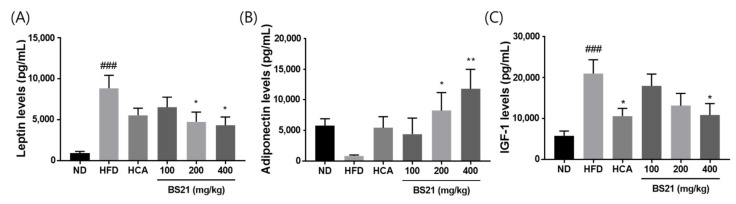
Effects of BS21 on serum adipokine levels. (**A**) Leptin, (**B**) adiponectin, and (**C**) IGF-1 concentrations. BS21, HFD + mixture of *Phyllostachys pubescens* leaf and *Scutellaria baicalensis* root extracts at a ratio of 2:1; HCA, HFD + *Garcinia cambogia* extract; ND, normal diet. Values are means ± SEM (*n* = 10). ### *p* < 0.001 vs. the ND; * *p* < 0.05and ** *p* < 0.01 vs. the HFD.

**Figure 4 nutrients-14-00446-f004:**
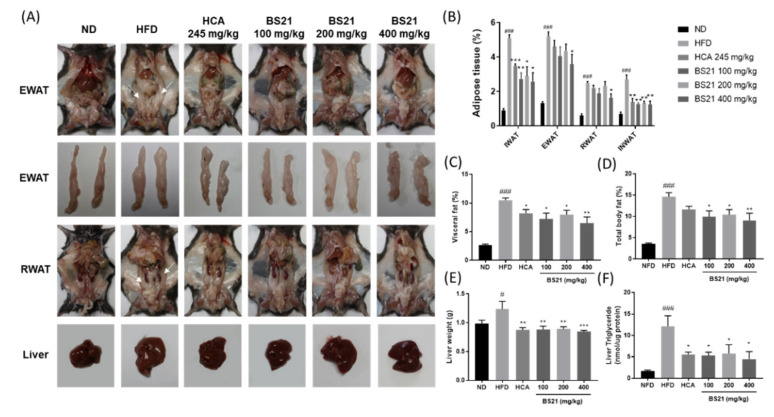
Effects of BS21 on adipose tissue and liver weights in high-fat diet (HFD)-induced obese mice. (**A**) A representative image for each mouse group. (**B**) Adipose tissue, (**C**) visceral fat, (**D**) total fat percentage, (**E**) liver weight, and (**F**) liver triglyceride levels. EWAT, epididymal white adipose tissue; INWAT, intestinal white adipose tissue; IWAT, inguinal white adipose tissue; RWAT, retroperitoneal white adipose tissue. Visceral fat includes EWAT, RWAT, and INWAT. Total fat includes subcutaneous fat (IWAT) and visceral fat. BS21, HFD + mixture of *P. pubescens* leaf and *S. baicalensis* root extracts at a ratio of 2:1; HCA, HFD + *Garcinia cambogia* extract; ND, normal diet. Values are means ± SEM (*n* = 10). # *p* < 0.05 and ### *p* < 0.001 vs. the ND; * *p* < 0.05, ** *p* < 0.01 and *** *p* < 0.001 vs. the HFD.

**Figure 5 nutrients-14-00446-f005:**
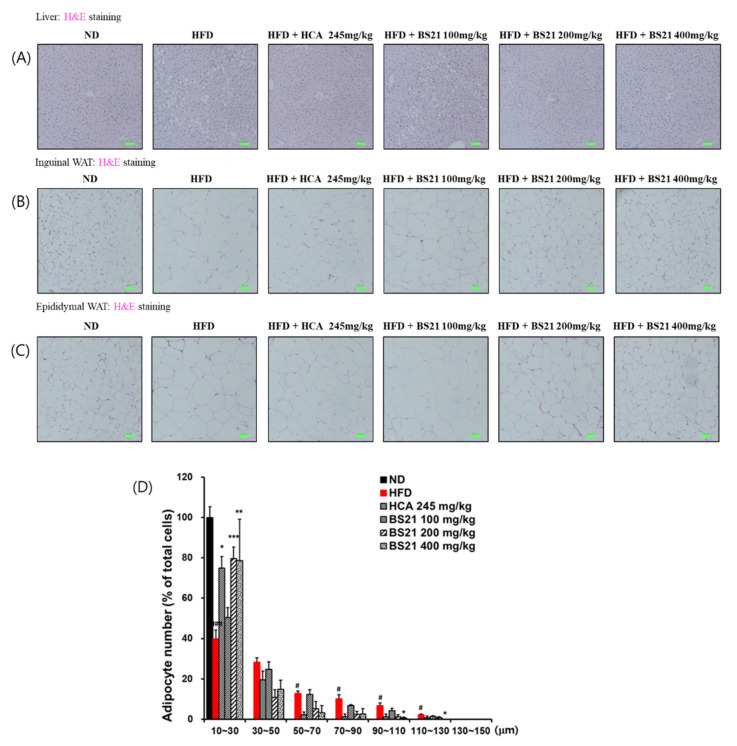
Liver and adipose tissue histology and adipocyte size in high-fat diet (HFD)-induced obese mice. (**A**) Liver, (**B**) inguinal white adipose tissue (WAT), (**C**) epididymal WAT (EWAT) histology (scale bar: 50 μm, green color; magnification: ×200), and (**D**) adipocyte size from EWAT. ND, normal diet; BS21: HFD + mixture of *Phyllostachys pubescens* leaf and *Scutellaria baicalensis* root extracts at a ratio of 2:1; HCA: HFD + *Garcinia cambogia* extract. Datas are means ± SEM (*n* = 10). # *p* < 0.05 and ### *p* < 0.001 vs. the ND; * *p* < 0.05, ** *p* < 0.01, and *** *p* < 0.001 vs. the HFD.

**Figure 6 nutrients-14-00446-f006:**
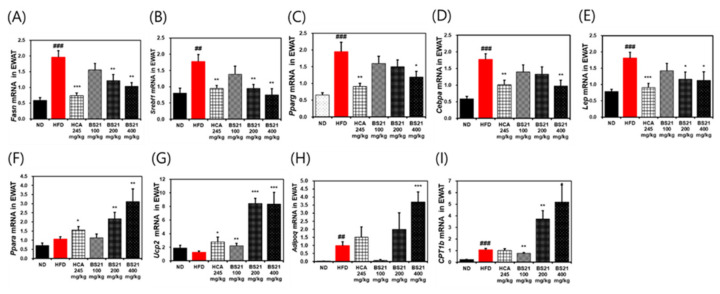
Effects of BS21 on lipogenesis (adipogenesis) or lipolysis-related gene expression in the white adipose tissue of high-fat diet (HFD)-induced obese mice. (**A**) *Fasn*, (**B**) *Srebf1*, (**C**) *Pparg*, (**D**) *Cebpa*, (**E**) *Lep*, (**F**) *Ppara*, (**G**) *Ucp2*, (**H**) *Adipoq*, and (**I**) *CPT1b* expression. The mRNA expression levels are expressed as fold increases relative to the HFD group after normalization to *Gapdh*. BS21, HFD + mixture of *Phyllostachys pubescens* leaf and *Scutellaria baicalensis* root extracts at a ratio of 2:1; HCA, HFD + *Garcinia cambogia* extract; ND: normal diet. Values are showed as the means ± SEM (*n* = 10). ## *p* < 0.01 and ### *p* < 0.001 vs. the ND; * *p* < 0.05, ** *p* < 0.01 and *** *p* < 0.001 vs. the HFD.

**Figure 7 nutrients-14-00446-f007:**
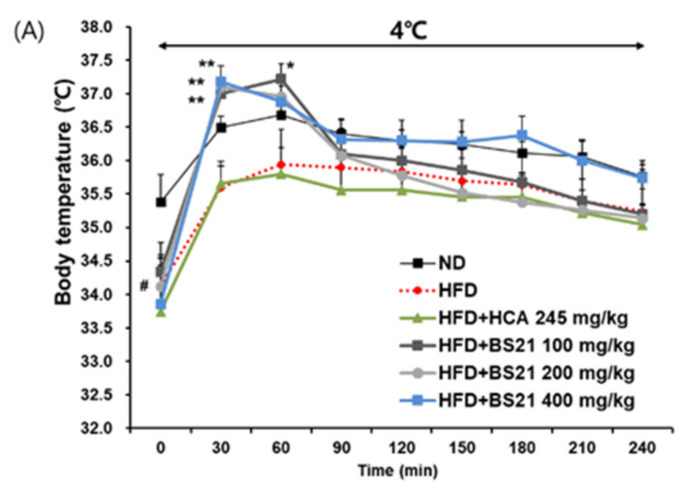
Effect of BS21 on cold-stimulated adaptive thermogenesis in high-fat diet (HFD)-induced obese mice. (**A**) Cold tolerance test and (**B**) UCP1 expression in brown adipose tissue using immunofluorescence (scale bar: 10 μm, white color; magnification: ×200). UCP1 and p-AMPK expression (**C**) in epididymal (EWAT) and (**D**) inguinal white adipose tissues (IWAT) using Western blot. BS21, HFD + mixture of *Phyllostachys pubescens* leaf and *Scutellaria baicalensis* root extracts at a ratio of 2:1; HCA, HFD + *Garcinia cambogia* extract; ND, normal diet. Values are means ± SEM (*n* = 10). # *p* < 0.05 and ## *p* < 0.01 vs. the ND; * *p* < 0.05 and ** *p* < 0.01 vs. the HFD.

**Figure 8 nutrients-14-00446-f008:**
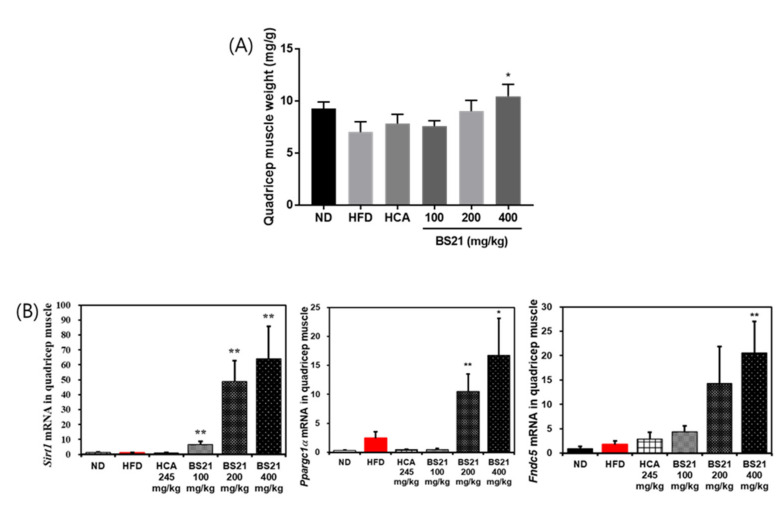
Effects of BS21 on energy metabolism-related gene expression in the muscle of high-fat diet (HFD)-induced obese mice. (**A**) Quadriceps femoris muscle weight and (**B**) mRNA expression of *Sirt1*, *Ppargc1a*, and *Fndc5* genes. Expression is indicated as the fold increase relative to the HFD mice after normalization to internal control *Gapdh*. BS21, HFD + mixture of *Phyllostachys pubescens* leaf and *Scutellaria baicalensis* root extracts at a ratio of 2:1; HCA, HFD + *Garcinia cambogia* extract; ND, normal diet. Values are showed as the means ± SEM (*n* = 10). * *p* < 0.05 and ** *p* < 0.01 vs. the HFD.

## Data Availability

The data presented in this study are available in this article.

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
