# Peer review of "Combined Phyllostachys pubescens and Scutellaria baicalensis Prevent High-Fat Diet-Induced Obesity via Upregulating Thermogenesis and Energy Expenditure by UCP1 in Male C57BL/6J Mice"

_nutrients, 2022, doi:10.3390/nu14030446_

Round 1
Reviewer 1 Report
The article titled: “Combined Phyllostachys pubescens and Scutellaria baicalensis prevent high-fat diet–induced obesity via upregulating thermo-genesis and energy expenditure by UCP1” is a strong thorough investigation of hyllostachys pubescens leaf and Scutellaria baicalensis root mixture extracts on obesity and obesity related markers in mice. While the article is well presented, I recommend the following changes to improve the article:
- A more robust introduction on the plants used would be helpful. The authors make general citations but do not discuss any specific studies using these plants.
- The title should reflect the model in which the study was conducted. In this case, the title should clearly state that these effects were studied in male C57BL/6J mice.
- What was the composition of the normal diet?
- Was the extract used in the entire study made on the same day? How was it stored? Did the authors make fresh extract throughout the study? This information is very important.
- Why were 245mg/kg chosen for HCA?
- Why did the authors choose not to have a group of normal diet plus BS21? Were these studies done previously? It would have been interesting to see the effect of BS21 on normal diets.
- For figure 4B, wouldn’t bodyfat percentage be a much better way to measure lipid weight? There must have been error introduced in this method.
- The order of materials and methods is not the same as the order of the results presented. For example, cold tolerance test is presented after histological analysis but appears first before histological analysis in the methods. This should be fixed so the reader can look at the data and the methods easier.
- Many methods are lacking in detail. For ELISA kits, the actual kits should be described, including catalog numbers. For anti-bodies, the clone # should be disclosed. More detail on how the data was analyzed would be helpful. For example, in Liver TG measurement, the authors just stated which kits were used, more details would be helpful.
- Immunofluorescence protocols are lacking. The wavelengths used, the microscope... many important details are just missing.
- The discussion could also be improved with more references to previous works in this area. The authors discuss the genes but not the work done with these plants. Do these results compare with previous findings using these plants? What gene changes were associated in previous studies with these plants?
Overall, I think the data is presented well, the authors should expand on their introduction, methods, and discussion.
Most importantly, the details in the methods must be improved.
Reviewer 2 Report
Comments to authors:
Sung et al. investigated the antiobesity benefits of a Phyllostachys pubescens leaf and Scutellaria baicalensis root combo (BS21) in a high-fat diet (HFD)–obese mice model, as well as the underlying mechanisms of action. BS21 lowered body weight, white adipose tissue (WAT), and liver weights, as well as hepatic lipid accumulation and adipocyte size. Furthermore, BS21 lowered non-esterified fatty acid, triglyceride, glucose, lactate dehydrogenase, low-density lipoprotein cholesterol, total cholesterol, leptin, and insulin growth factor 1 serum concentrations while increasing adiponectin. In WAT, BS21 reduces the gene expression of lipogenesis-related proteins (PPAR, SREBP-1c, C/EBP-, fatty acid synthase, and leptin), while increasing the expression of lipolysis-related proteins (PPAR-, UCP2, adiponectin, and CPT1b). Cold-stimulated adaptive thermogenesis and UCP1 protein expression with AMPK activation in BS21 treated mice adipose tissue, as well as enhanced browning of WAT and mRNA expression of energy metabolism-related proteins SIRT1, PGC-1, and FNDC5/Irisin in the quadricep muscle. The authors concluded that BS21 had anti-obesity and antihyperlipidemic effects in HFD-induced obese mice via boosting thermogenesis and energy expenditure and regulating lipid metabolism.
Overall, the manuscript was well written, and the study was well-executed with adequate experiments; nonetheless, the following points should be clarified:
Comments:
- Similar kinds of studies were published by the same group of authors with various dosages of BS21 utilized; appropriate justification for the uniqueness and the supplied doses of BS21 used in this study must be provided.
- What was the reasoning behind the 2:1 ratio of Phyllostachys pubescens leaf and Scutellaria baicalensis root combination (BS21)?
- Mice were given an HFD with BS21 (100, 200, or 400 mg/kg) for 9 weeks; does 9 weeks of HFD with BS21 or HFD began sooner make a difference?
- Given that this is a diet-induced obesity model that also promotes insulin resistance, fatty liver, and so on, what were the blood glucose levels in the control and experimental groups? This is something that has to be addressed in the updated paper.
- 1What is the reason for employing these mRNA expressions SIRT1, PGC-1, and FNDC5/Irisin in the quadricep muscle for energy metabolism-related proteins? And why did the authors choose the quadriceps muscle above others; a rationale must be supplied.
- "The quantities of adipokines and insulin growth factor (IGF) 1 hormone in serum were measured using enzyme-linked immunosorbent assay (ELISA) kits (R&D Systems, Minneapolis, MN, USA)." Please provide a catalog number for these kits. Are these kits designed for mice or do they have cross-reactivity?
- In the revised manuscript, histology and immunostaining scale bars and magnifications must be included.
- Figure 7 C loading control is not equal; provide representative blot.
